# Influence of Underwater Bar Location on Cross-Shore Sediment Transport in the Coastal Zone

**Olga Kuznetsova** [1,2,*] **and Yana Saprykina** [2]

1   Zubov State Oceanographic Institute of Roshydromet, Russian Academy of Sciences, Moscow 119034, Russia
2   Shirshov Institute of Oceanology, Russian Academy of Sciences, Moscow 117997, Russia; saprykina@ocean.ru
*   Correspondence: olga.ku-ocean@yandex.ru; Tel.: +7-903-557-9035

**Abstract:** The effect of the underwater bar position on a sandy beach profile was studied on a timescale of one storm, using the XBeach numerical model. The largest shoreline regress occurred in the first hour of storm. For the chosen wave regime an underwater profile close to the theoretical Dean's equilibrium profile is formed after 6 h. The position of the underwater bar affects the shoreline retreat rate. The lowest shore retreat occurs when the bar crest is located at a distance equal to 0.70–0.82 of the deep-water wavelength, corresponding to the period of the wave spectrum peak. The maximal shoreline retreat occurs when the bar is located at a distance that is close to a half wavelength. The shoreline recession depends on the heights of low-frequency waves. The smaller the mean wave period and the higher low-frequency waves' height near the coast, the smaller the retreat of the shoreline. The distance of seaward sediment transfer is directly proportional to the significant wave height near shore.

**Keywords:** coastal zone; storm deformations; underwater bar; XBeach; wave transformation; cross-shore sediment transport; equilibrium profile

## 1. Introduction

Hydrodynamic processes are important factors in coastal zone evolution. The off- and onshore relief of sandy beaches is deeply bound with the wave regime. The wave climate and its variations are main mechanisms of cross-shore sediment transport in the coastal zone; for instance, the formation and movement of underwater longshore bars, which are observed on many sandy coasts [1–3].

Around 10% of sea coasts have underwater bars [4]. The timescale of longshore bar formation and movement can vary from days to months [5]. According to laboratory experiments [3], under weak or moderate waves, the underwater bar moves shoreward until it joins the coast and disappears. Stronger waves switch the direction of the bar movement seaward. With changing wave conditions, the underwater bar can stay approximately at the same place and be considered stable [3].

Underwater bars are specific features of the bottom relief, so that they affect a wave transformation process within a coastal zone. As a result, the bars have influence on a cross-shore sediment transport and shoreline deformations. The wave transformation over barred profiles of sandy beaches and the corresponding morphodynamical features are a challenging and intensively studied topic [1–3,6].

Nevertheless, the role of underwater bar positioning in shoreline dynamics is still not obvious. From an engineering point of view, this issue is important for coastal defense, and it should be clarified [7]. Artificial underwater bars and reefs that imitate natural structures have become popular in coastal engineering [8]. Such constructions (breakwaters) are installed in order to decrease the wave load on the coast and reduce erosion. The decline of wave energy occurs due to breaking and shortening of the mean wave period by non-linear dispersive wave transformation over bars [9].

A similar effect is also detected in studies devoted to the impact of wave farms on nearshore wave conditions and coastal protection [10]. It is crucial to find an appropriate position and an optimal shape for artificial underwater structures, in order to obtain the maximum benefits. Thus, a detailed study of the influences of the bar position on wave transformation, the corresponding sediment transport in the coastal zone, and the rate of wave-induced shoreline erosion, is a very important scientific and coastal engineering task.

The goal of this work is to investigate the influence of the underwater bar position (off a non-tidal sea coast) on the transformation of waves above it, and on corresponding cross-shore sediment transport, on the timescale of a strong storm.

## 2. Materials and Methods

During the field experiments at the Shkorpilovtsy study site (Black Sea, Bulgaria) [11], we noticed the festoon-shaped features of the shoreline. Satellite images show that the onshore festoon-pattern is accomplished by crescent underwater bars (Figure 1). According to long-term observations [11,12], the bars migrate slightly, depending on variations in the wave conditions. The shoreline shape has inter-annual variations that are possibly associated with the features of the underwater bottom relief. This fact induced us to prove the idea that the location of the underwater bar defines the shape of shoreline to some extent.

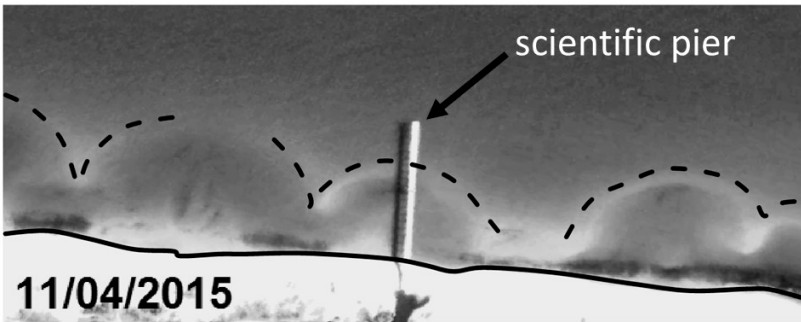

**Figure 1.** Satellite image of the crescent underwater bars at the Shkorpilovtsy study site (Image © 2017 Digital Globe). Dotted line: the crest of the bar; solid: the water's edge.

Numerical computation by using XBeach [13] has been chosen as a main tool, because this is a well-developed and popular open-source hydrodynamic and morphology modelling package [14,15]. The non-hydrostatic mode of XBeach, which we used for wave and bottom changes modelling, resolves short waves and provides an accurate reproduction of wave propagation in shallow water [16]. In contrast to stationary or surf-beat mode, the non-hydrostatic mode resolves the wave profile, and it does not require additional wave asymmetry correction.

Although the main goal of the research involves a purely numerical experiment, it is still important to set reasonable realistic boundary conditions. For this purpose, we used field observations (bathymetry and sediment properties) from the Shkorpilovtsy study site [17], as well as information about the wave climate in the north-western part of the Black Sea [18,19].

A numerical grid (1D) was built, based on a set of 12 cross-shore profiles, measured in a frame of the international field experiment "Shkorpilovtsy-2007". All of the observed profiles that were made along the beach had a bar that was located at different distances from shore. From the observed data, we obtained a characteristic shape of the bottom profile and the underwater bar.

The average bottom profile on the Shkorpilovtsy coast has no bar. It has a slope of 0.022, a slight increase of the slope in the upper part, and a small terrace at 2–3 m depth. This profile was considered to be used for modelling, and it was a basis for the creation of a set of barred profiles.

The characteristic shape of the underwater bar was superimposed with a mean profile at a different distance from the shoreline. Thus, five synthetic profiles were created with different bar positionings (Table 1, Figure 2).

**Table 1.** Parameters of the initial profiles.

| Parameter | Profile | | | | | |
|---|---|---|---|---|---|---|
| | 0 | 1 | 2 | 3 | 4 | 5 |
| Depth over the bar crest, m | None | −3.07 | −2.68 | −2.38 | −2.20 | −2.08 |
| Bar crest location (x-coordinate), m | None | 714 | 730 | 748 | 762 | 784 |
| Distance between shoreline and the bar crest (X), m (distance (X)/wave length (L)) | None | 141 (0.82) | 125 (0.72) | 107 (0.62) | 93 (0.54) | 73 (0.42) |

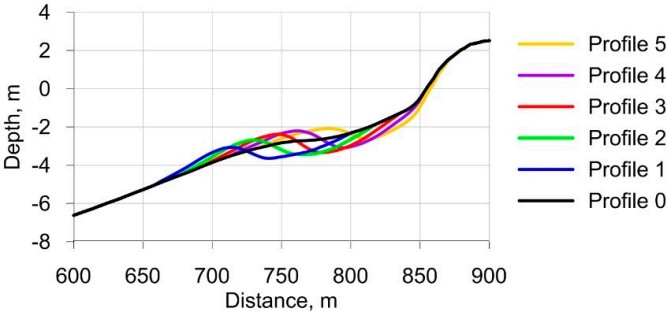

**Figure 2.** The cross-shore profiles used as the model bathymetry input.

The spatial resolution of the grid was set to 2 m, as calculated using Matlab Toolbox, which has been created and recommended specifically for this purpose, according to the XBeach developers. The Toolbox helps XBeach users to choose appropriate grid settings, taking into account the wave parameters and the relief characteristics. The Black Sea is a non-tidal sea, and so the initial water level was set at 0 m for all of the simulations.

The sediment on the Shkorpilovtsy beach are anisogamous quartz sands. More than 95% of the bottom sediments in the upper part of profile (till 2.5 m) are coarse-grained or medium. For the modelling, we used medium grain, with a diameter D50 (d50) of 0.2 mm.

The validation of XBeach, which has been made by developers and users, shows that this package can be successfully used in non-hydrostatic mode with the default settings [20]; however, some studies have shown that XBeach overestimates coastal erosion [21], which we also noticed from our analysis of the field data and the modelling results. For study site conditions, we tested XBeach with stationary and non-stationary (Joint North Sea Wave Project Spectrum - JONSWAP) wave inputs, and compare these with the synchronous observations carried out during field experiment <Skorpilovtsy-2007> (Figure 3). The results could be considered reasonable.

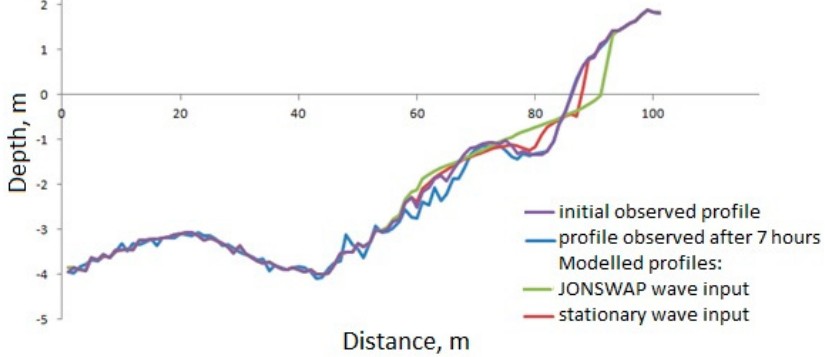

**Figure 3.** Results of the XBeach verification for the Shkorpilovtsy study site, for cases of 7 h wave action (significant wave height Hs = 0.9 m, and spectral peak period Tp = 7 s).

The goal of the research was to investigate the wave-bottom adjustment on the time scale of one storm. For this reason, we defined the wave input in accordance with typical storm conditions, presented in the wave climate descriptions of Black Sea [18,19]. The wave input was set at the sea boundary of the numerical grid ($\approx$860 m from shore) in the form of JONSWAP wave spectra, with the following parameters: peak enhancement factor $\gamma = 3.3$, significant wave height Hs = 1.5 m, spectral peak period Tp = 10.5 s, wave energy-spreading angle $\delta = 2.5°$. A storm with wave heights of 1.5 m was considered to be a dangerous hydrometeorological phenomenon in the Azov–Black Sea region [22]. A wave period of 10.5 s corresponded to extreme storms with return periods of 25 years [18]. The duration of a single storm event was set to 20 h, in accordance with the criteria developed for the Black region by Belberov [23], and for Atlantic coasts by Lozano [24], also taking into account the World Meteorological Organization recommendations for meteorological observations [25].

The modelling process was organized according to the following algorithm. In first hour of wave action, there was no morphology changes, but there was a wave output with very fine time resolution (5 Hz). Modelling was continued for another 20 h, with relief changes enabled. The output of the bathymetry was set every hour.

From the first step of the computation, we obtained free surface elevation data along the profile between the coordinates 620–820 m (see Figure 4). This part corresponded with depths of 2–6 m. The sampling frequency of the time series was 5 Hz. Chronograms were used for the calculation of the wave spectra, and for the following wave parameters [26]:

1. Significant wave height (in m), calculated as:

$$H_S = 4 \cdot \sqrt{m_0} \tag{1}$$

where:

$$m_0 = \int_0^\infty S(\omega)d\omega \tag{2}$$

and $S(\omega)$ are the spectra, $\omega$ is the angular frequency, with linear frequency filters: 0–0.05 Hz chosen to account for a significant wave height of low frequency range, including infragravity waves ($H_{IGW}$).

2. The mean wave period(s) is/are as follows:

$$T_{mean} = \frac{\int_0^\infty S(\omega)d\omega}{\int_0^\infty \omega \cdot S(\omega)d\omega} \tag{3}$$

3. The wave asymmetry coefficient (with asymmetry relative to the vertical axis):

$$As = \frac{\langle \zeta_H^3 \rangle}{\sigma^3} \tag{4}$$

where $\langle \rangle$ is the averaging operator, $\zeta$ are the free surface elevations, $\sigma$ is the standard deviation of the free surface elevations, and $\zeta_H$ is the Hilbert transform.

4. The wave-skewness coefficient (relative to the horizontal axis):

$$Sk = \frac{\langle \zeta^3 \rangle}{\sigma^3} \tag{5}$$

From the second step of computation, we obtained the hourly data of the computed morphology changes for over 20 h of wave action. Based on a set of calculated profiles, the coastline retreat and the change in the underwater bottom relief were evaluated.

The parameters of the underwater bar for the initial profiles (input conditions) are shown in Table 1. The wave length was determined by the dispersion relation of the linear waves' theory for the spectral peak period.

## 3. Results

Figure 3 shows the underwater bottom profiles, both initially and the result of 20 h storm modelling. The resulting deformations of the underwater profile are presented in Table 2. The change in all of the beach profiles was characterized by erosion in the splash area, the transition of material seaward, and its accumulation at depths of more than 1 m. However, the activity of erosion and accumulation processes on profiles with different bar locations varied.

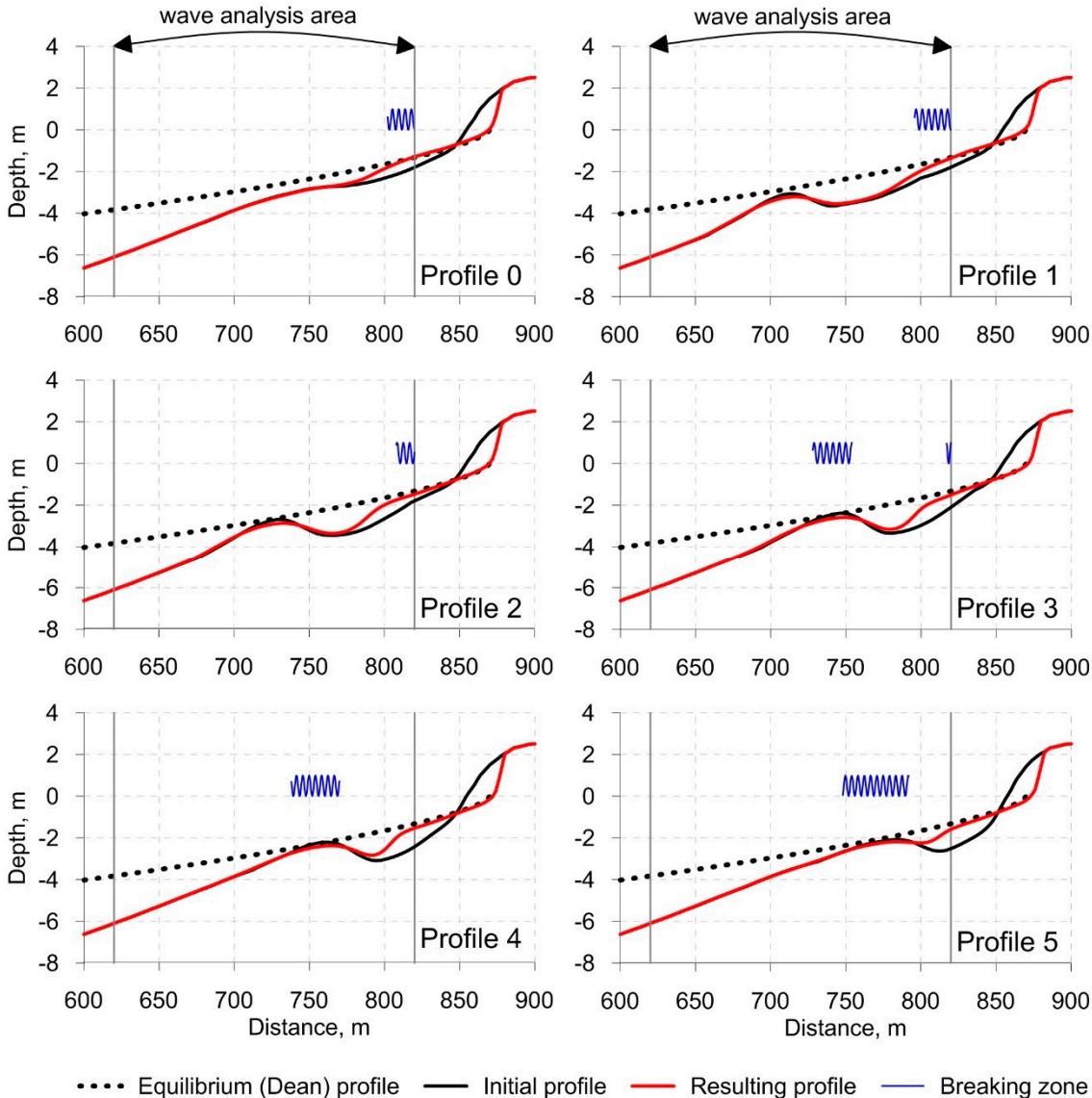

**Figure 4.** Initial and resulting profiles after 20 h of wave action (Hs = 1.5 m, Tp = 10.5 s), location of the breaking zone, and theoretical shape of the equilibrium profile calculated for a grain size of 0.2 mm.

**Table 2.** Main changes in model profiles after 20 h of wave action.

| Parameter | Profile | | | | | |
|---|---|---|---|---|---|---|
| | **0** | **1** | **2** | **3** | **4** | **5** |
| Depth over the bar crest, m | None | −3.21 | −2.86 | −2.59 | −2.37 | −2.18 |
| Distance between the shoreline and the bar crest (X), m (distance (X)/wavelength (L)) | None | 141 (0.82) | 125 (0.72) | 107 (0.62) | 93 (0.54) | 73 (0.42) |
| Shoreline regression (Sh), m | 14.8 | 14.3 | 15.3 | 16.7 | 17.6 | 15.6 |
| Average accretion layer, m | 0.34 | 0.29 | 0.30 | 0.41 | 0.50 | 0.61 |
| Distance of sediment transportation (x), m (Distance of sediment transportation (x) / Wavelength (L)) | 76 (0.44) | 86 (0.50) | 98 (0.57) | 82 (0.48) | 70 (0.41) | 54 (0.31) |

In general, if a bar is located closer to the coast and it has reduced depth above it, several patterns can be distinguished: the rate of shore erosion increases, the thickness of the sediment accumulation layer in the underwater part of the profile decreases, and the distance of sediment transfer to the sea grows (Table 2). However, the shoreline recession (Figure 5a) on the profile with the furthest bar position (0.82 from the wavelength) was substantially less than that of the profile where the bar was located closer to the shore (0.42 from the wavelength). The coastline degradation on the profile where the bar was positioned at 0.54 from the wavelength was maximum for all of the considered profiles. The seaward transfer distance of the sediment (Figure 5b) was maximal on the profile where the bar is positioned, at 0.82 (profile 1) from the wavelength. In all cases, there was no transport of sediments beyond the bar (Figure 4).

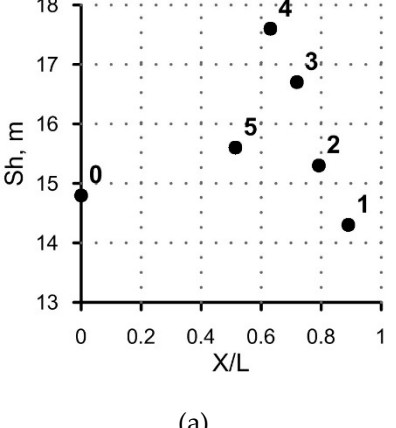
(a)

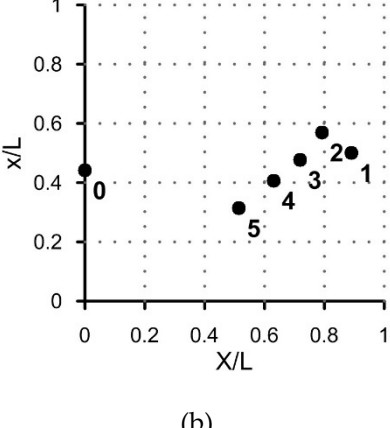
(b)

**Figure 5.** (**a**) Shoreline regression (Sh, m) and (**b**) distance of seaward sediment transport, in terms of wavelength (x/L) independence of the initial bar location (X/L). X is the distance between the bar crest and the shoreline, x is the distance of the maximum amounts of sediments that are transported away from the shore, L is the wavelength calculated for deep-water conditions

Compared to the profile without a bar, the profile with the furthest bar location (the relative distance from the coast is 0.82 of wavelength) reduces the degree of shoreline degradation. The profile with the closest position to the bar (a relative distance from the coast of 0.42, or less than half the wavelength) worked best as a barrier against carrying the beach material seaward to depth. Compared with a barless control profile, an underwater bar at some relative distances from the coast could lead to an increase in coastline degradation.

Under the storm waves, slight deformations of bars also occurred (Figure 4): the reduction of its relative height, the deepening of the bar top (on profile 5), and a retreat of the bar crest (2–4 m) towards the sea (on profiles 1, 2, and 4). The bar shape changed with regard to its slope oriented towards the

coast, which became less steep due to the filling of the bar trough. Such movements of the bar crest and the change in its asymmetry are generally consistent with the data of field observations [17].

The process of the retreat of the coastline and the transformation of the coastal zone relief occurs non-uniformly over time. Figure 6 shows the changes in the coastline retreat speed after 10 h of wave action, for profiles with maximum (4) and minimum (1) changes. The fastest shoreline retreat was observed during the first hour of the storm for all profiles: it varied from 4.5 to 6.5 m/h. Erosion activity slows down over time. After 6 h of wave action, the beach profile adapts to the specific waves occurring, and assumed a relatively equilibrium state. The shoreline regression rate became ≈0.5 m/h, and it remained approximately the same for all profiles.

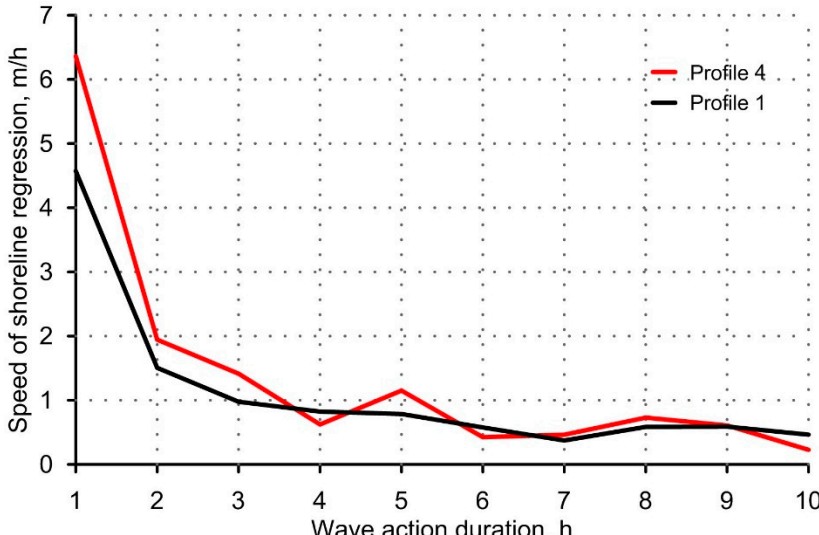

**Figure 6.** The speed of shoreline regression (m/h) variations during the numerical experiment that was run on profiles 1 (black) and 4 (red).

Regardless of the initial underwater relief, an underwater terrace was is formed on for all profiles (Figure 4), representing an equilibrium profile that was close to the theoretical classical Dean's profile proposed in [27]:

$$h = Ax^{2/3} \tag{6}$$

where $A = 0.1$, calculated for a sediment grain size of 0.2 mm [26]. The formation of an underwater profile of similar shape with a terrace under the influence of uniform storm waves was also observed by us in the field experiment "Shkorpilovtsy 2007" [17].

Different degrees of coastline degradation during the first hour of the storm can be explained by differences in the transformation of the storm waves along the bottom profiles. Figure 7 shows the dependencies of the change in the coastline retreat, and the sediment seaward transfer distance on the relative change in the significant wave heights, as determined by the ratio of the corresponding values before and after the bar (with coordinates on model profiles 620 and 820 m). The significant wave height slightly decreases when the waves came nearer to the shore, but the decline was more strongly expressed in profiles 4 and 5, where the bar is located near the shore. On profiles 1–3, the significant wave height fall was less than in the profile without a bar. The distance of seaward sediment transport directly depends on the significant wave height: the greater the height, the further the material is transferred (Figure 7b). The relation between the shoreline regression and the change in significant wave height was not so clear (Figure 7a).

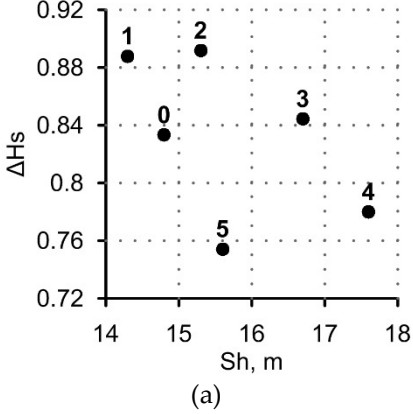 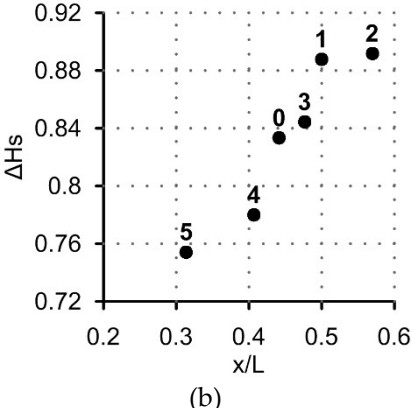

**Figure 7.** Dependence of (**a**) the shoreline retreat (Sh, m) and (**b**) the relative distance of seaward sediment transport (x/L) on the relative change in significant wave height (ΔHs, m) occurring during wave transformation over barred profile (620–820 m, see Figure 4). L is wave length calculated for deep water conditions, x is distance of maximal sediments transport away from the shore

The shoreline retreat is influenced more by the mean wave period than by wave height. Figure 8 depicts the relation between bottom deformations and relative change of mean wave period determined by the ratio of the corresponding values before and after the bar (coordinates on model profile 620 and 820 m). The smaller the change in the mean wave period, the smaller the degradation of the coastline (Figure 8a). Such a change of parameters occurred in profile 1, with a bar being located at a distance from the shoreline of 0.82 of a wavelength. There was no clear dependence of sediment transport on the mean period.

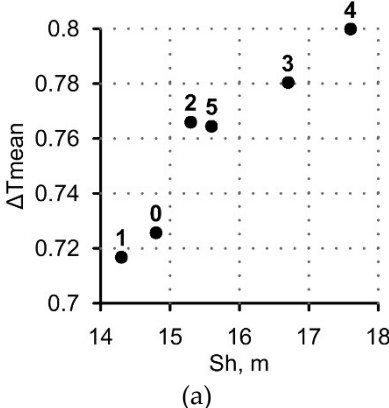 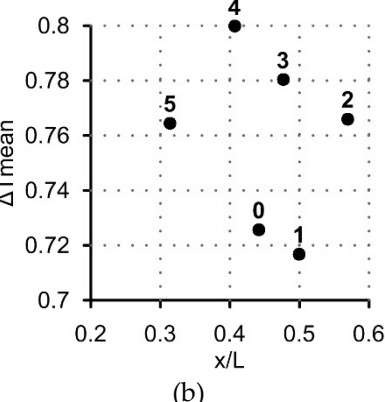

**Figure 8.** Dependence of (**a**) shoreline retreat (Sh, m) and (**b**) the relative distance of seaward sediment transport (x/L) on the relative change in the mean wave period (ΔTmean, s) occurring during wave transformation over a barred profile (620–820 m, see Figure 4). L is the wavelength calculated for deep-water conditions, and x is the distance of maximal sediment transportation away from the shore.

The modelling of profiles with five different bar locations shows that a growth of significant height in the low-frequency waves after passing a bar leads to a decline of coastline degradation (Figure 9a). This relation is close to linear, except for profile 5, where the change in the low-frequency wave height is the same, but the shoreline regression rate and the distance of seaward sediment transport (Figure 9b) are different. This could be caused by the different natures of low-frequency waves. Waves of low-frequency bands can include infragravity waves (IGW) of different kinds: bound long waves and break point-forced long waves [6]. Previous investigations have shown show that various IGW affect the coast in two different ways: a) bound long waves protect the shore, b) the break-point forces long waves, leading to an intensification of near-shore erosion [28].

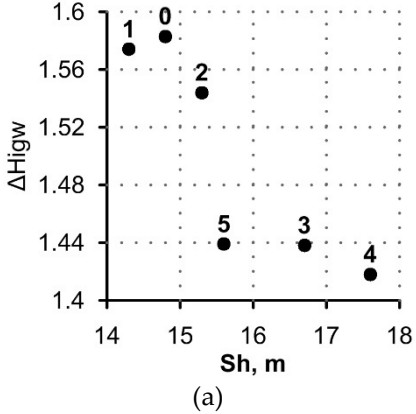 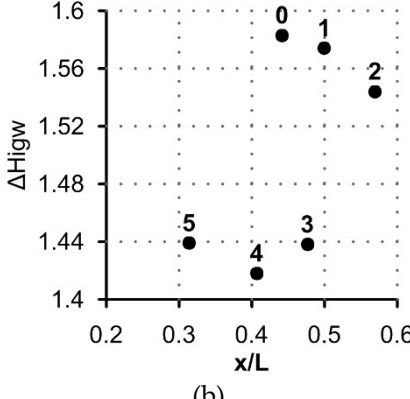

(a)        (b)

**Figure 9.** Dependence of (**a**) the shoreline retreat (Sh, m) and (**b**) the relative distance of seaward sediment transport (x/L)over a on relative change in mean wave period (ΔTmean, s) occurring during wave transformation over a barred profile (620–820 m, see Figure 4). L is the wavelength calculated for deep-water conditions, x is the distance of the maximal sediments transported away from the shore.

The presence of the underwater bar changed the symmetry of the waves. The wave skewness, after passing the bar (the x-coordinate on the profile was 820 m), remained almost similar ≈ 1.6 for all profiles. High values of the skewness coefficient show that breaking in XBeach model is probably described as spilling, because in observations and laboratory experiments plunging breaking waves have skewness less than 1 [29]. Waves breaking by spillage are symmetric relative to the vertical axis, they have sharp crests and flat troughs, while plunging breaking waves have steep fronts [30].

The wave asymmetry coefficient behaves differently. Figure 10 shows the wave asymmetry coefficient in a nearshore area (the 820 m coordinate on the profile) and its relation with shoreline regression and seaward sediment transport distance. According to the model data, the more asymmetric waves are after the bar, the less the coastline degrades.

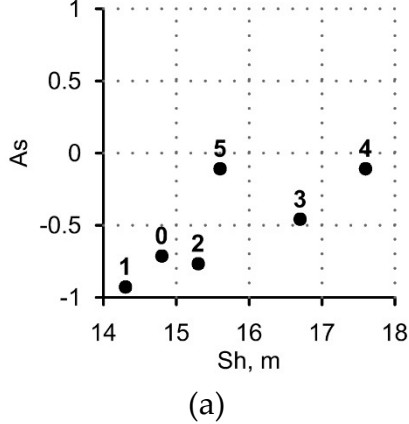 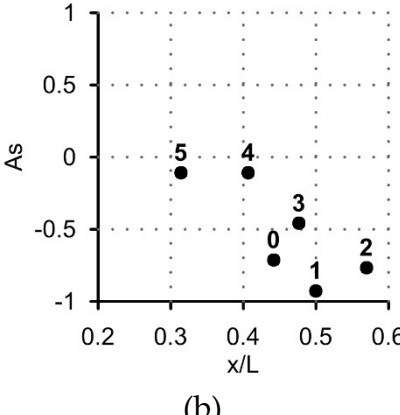

(a)        (b)

**Figure 10.** Dependence of (**a**) shoreline retreat (Sh, m) and (**b**) the relative distance of seaward sediment transport (x/L) on wave asymmetry (As) after wave transformation (near the shore, at the x-coordinate 820 m). L is the wavelength calculated for deep-water conditions, and x is the distance of maximal sediments transported away from the shore.

The most asymmetric waves behind the bar were observed on profiles where the first wave breaking occurred, between the coast and the bar, for example, on profiles with the bar's distant location (profiles 1 and 2), as well as on a profile without a bar (profile 0, see Figure 3). When the bar was located closer to the shore, and accordingly, the depth decreased above it, the waves broke both at the top of the bar, and near the shore. In this case, the waves broke over the bar farther from the shore than on the profiles 0–2, and the second breaking zone was closer to the coastline. The presence of

two breaking points leads to a significantly greater shoreline regression. The exception for this is the profile 5, where the bar is located at the closest distance to the coast line. The bar prevents seaward sediment transport, so that the shoreline degradation is reduced.

An Explanation of wave asymmetry impact on sediment transport w discussed in detail in [30]. Cross-shore sediment transport is defined by the balance of wav- induced sediment transport directed to the shore, and the undertow, which moves sediments seaward. According to Bailard's formulation of wave-induced sediment transport discharge [31] depends on highest statistical moments of near bottom velocity:

$$q = \frac{1}{2} f_w \rho \left( \frac{\varepsilon_b}{\tan \Phi} \overline{u|u|^2} + \frac{\varepsilon_s}{W_s} \overline{u|u|^3} \right) \tag{7}$$

where u = u(t): the instantaneous near-bottom velocity.

Formula (7) was adopted by Leontiev [6] for calculations through amplitudes of first and second nonlinear harmonics of near bottom-velocity and cosines of phase lag between them (bi-phases):

$$\overline{u|u|^2} = \frac{3}{4} u_m^2 u_{2m} \cos \beta, \overline{u|u|^3} = \frac{16}{5\pi} u_m^3 u_{2m} \cos \beta \tag{8}$$

where $u_m$ and $u_{2m}$ – amplitudes of first and second harmonics, β- phase shift between the first and second nonlinear harmonics (bi-phases).

As it was revealed in [31], the wave asymmetry coefficient *A*s was linear, depends on bi-phase:

$$As = 0.8\beta \tag{9}$$

Thus, according to (8 and 9), the shoreward wave induced-sediment discharge depends on the cosine of the bi-phase, or As magnitude. Maximum of it will occur when bi-phase (and accordingly *A*s) is zero, because cosine will has maximal value. A decrease of wave-induced sediment discharge due to wave asymmetry will lead to increase in the role of the undertow in sediment transport. Therefore, sediments will move more seaward, and the shoreline will retreat more, due to the erosion in nearshore zones (see Figure 10b).

## 4. Conclusions

We carried out a study on the influence of the underwater bar location on the transformation of the waves above it, and on the corresponding cross-shore sediment movement in the coastal zone, on a time scale of one storm. This allows us to conclude on the following:

1. The position of the underwater bar affects the shoreline degradation and the distance of seaward sediment transport. The maximum transfer of sediments towards the sea is within the distance between the shoreline and the underwater bar crest. A minimum of sediment movement occurs when the bar is located away from the shore, at a distance of less than half the wavelength, in deep water. The coastline retreat is minimal if the bar is located away from the coast, at a distance of 0.7-0.82 from the wavelength in deep water. In these cases, the underwater bar will have a more protective effect on the shore, compared to a profile without the bar.

2. The presence of the underwater bar located at specific distances from the coast may lead to an increase in shoreline degradation. If there is a longshore underwater bar that is located at an angle to the coastline, the non-uniformity of the coastline retreat, and the formation of festoons are possible.

3. The greatest difference in coastal retreat rate associated with the underwater bar location is observed within the first hour of the storm. Regardless of the location of the underwater bar on the initial profile, the equilibrium profile is formed after 6 h, for the selected wave conditions. The resulting equilibrium profile contains an underwater terrace, and it is close to the classical equilibrium profile. At the same time, the erosion rate slows down significantly and becomes identical for all profiles.

4. Changing the parameters of the waves during their transformation over different profiles has a significant impact on the degree of transformation of the underwater beach profile. It has

been established that there is an inverse relationship between the retreat of the coast line and the low-frequency wave heights near the coast. The decrease in the mean wave period, which is associated with the growth of higher harmonics during the passage of waves above the bar, reduces shoreline erosion. The distance of seaward sediment transport transfer is directly related to the significant wave height.

5. When waves propagate over profiles with underwater bars that are located at different distances from the coast, the wave asymmetry changes differently. According to the modelling results, the increase in wave asymmetry near the shore due to the existence of the bar leads to a decrease in the influence of waves on the coastal retreat.

**Author Contributions:** Both authors took part in field works on Shkorpilovtsy study site. XBeach modelling was conducted by O.K. in frame of post-graduate study (Y.S. was a scientific advisor). Analysis of modelling data was carried out by O.K. and Y.S.

**Funding:** This research was performed in the framework of the state assignment theme No. 0149-2019-0005, and supported in part by RFBR project No. 18-55-45026.

**Acknowledgments:** Authors appreciate collaboration with colleagues from IO BAS (Varna, Bulgaria) and Zenkovich laboratory of the sea shelf and coasts (IO RAS, Moscow, Russia).

**Conflicts of Interest:** The authors declare no conflict of interest.

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
