# Peer review of "Influence of Underwater Bar Location on Cross-Shore Sediment Transport in the Coastal Zone"

_jmse, doi:10.3390/jmse7030055_

Round 1

Reviewer 1 Report

I found myself with a number of questions and concerns after reading this paper, which I will try to summarize here.  First, I don't think that a stand-along modelling study is necessarily a contribution, at least without considerable discussion of other studies that have actually validated the model chosen.  In this case there is no discussion of model validation, model skill at predicting the parameters presented in the paper, and no discussion of uncertainty.  The lack of this sort of discussion (or ideally, validation provided as part of the study) makes the results and conclusion presented lack credibility.  Next, I was confused by the brief discussion of a field site, that is somehow used as a basis for the profiles used.  However, there is absolutely no other discussion of the field site, no field validation presented...its very confusing.  Next, there is very limited discussion of the boundary conditions used for the modelling, but no justification of those boundary conditions outside of a brief mention in the text that they are considered extreme for the field site.  Does that matter?  Are those boundary conditions representative of other locations such that the results may be useful elsewhere?  Lacking that sort of context, its hard to understand the contribution that this model investigation may represent.  Next, there is little connection of the results to other investigations in the discussion - what have others found?  How is this a contribution?  There is some allusion in the introduction to the idea that the authors intend this to primarily be used as a guide for the design of certain types of breakwaters - if so that should be clearly stated, and discussed in the context of other literature.  Finally, in regards to broader implications there are two alternative theories presented about the formation of underwater bars, but the methods/results don't necessarily seem to relate to bar formation, and there is no follow up discussion of these two theories and how your results clarify which is supported. 

In regards to the methodology, I found myself confused about the boundary conditions used in the two separate modelling segments (wave evolution and morphology evolution). There is some implication in the results that a range of wave conditions were used, yet the text seems to suggest otherwise?  A table clearly specifiying the wave and water level boundary conditions would be useful.  Also in regards to methods, there is some reference in the text to infragravity waves at frequencies as low as 0.05 hz...but I generally think of infragravity waves as having MUCH lower frequencies than that.  Again, its also not clear how variations in boundary conditions, which are alluded to in the results, are applied in the model domain.

For results, I struggled to interpret the style of plot presented in Fig 4, 6 and 7, and find in those the patterns that I think the authors wanted me to find based on the text.  I would suggest presenting results in stand along plots, and maybe providing annotation or additional information in those figures to better support the results and discussion.  

For what it is worth, I am also providing a scan of a marked up copy of the draft paper.

Author Response

We thank Reviewer for the time and effort and we appreciate very much all the constructive comments. Each comment has been carefully considered and responded point by point in attached file.

Reviewer 2 Report

The authors of the manuscript have used X-beach model to simulate underwater bar position and how the waves are transformed as they go over the bar.  The study shows that the if the underwater bar is closer to the shoreline, then the retreat of the shoreline may be maximum. The simulations are done for strong storm conditions. This paper has the potential to be interesting for the community that is currently focused on this topic of cross-shore sediment transport modeling. However, it lacks depth in explaining some of the model results. In the introduction, the authors mention two reasons for the growth of the underbars but fail to connect the results and conclusions to these points. The authors donot provide full information of the xbeach model setup that includes tuning parameters for accounting for wave asymmetry (“Facua”).  The figure quality is poor and needs a lot of improvement for archival purposes. Figures can be split up for shoreline regression and sediment transport or just present one to explain the physics. Use legends to mark the dots and triangles for various profiles.  

Because the topic of research is of crucial importance to the community, the paper can only be accepted after major revisions are made.

Below are some of the specific comments that can improve the paper quality:

1.       Line 11  - xbeach numerical model

2.       Line 12  - remove “classical”

3.       Expand on the methods specially the model setup. What are the sediment properties ? How are

What are the Xbeach model to get the asymmetric sediment transport ?

4.       In line 104 you mention the important fact that the focus for the plots needs

to be from 2-6 m depth and from a distance between 620-820 m. I recommend using this for all the plots to get a zoomed in picture of the results.

5.       Remove the extra “s” in table 2

6.       Why is the Dean’s profile in grey color in figure 3 profile 3.

7.       Lines 129-133 - I will refer to each result in context of the profile. Sediment seaward transfer distance is same where bar is positioned at 0.54 from wavelength and 0.82 from wavelength? I could not see this in table 2. Are you talking about profiles 1 and profiles 4 ?

8.       Lines 139-142 – This result was mentioned in the previous paragraph and the same text is getting repeated.

9.       Improve figure 4 and separate the two results of shoreline regression and sediment transport distance (Make fig 4a and 4b). Also, use legends to mark the dots and triangles in the figures.

10.   Lines 142-149 These recommendations on the underwater bar arrangement should be made in the discussion or summary section. This section should solely focus on the results.

11.   The sentence that talks about the angle of the bar relative to the coastline is out of context, there is no context about the angle of the coastline prior to this.

12.   Line 151- “deepening”

13.   Line 152 -154 – Can you plot these profiles and use arrowheads on the plot to mark

What you are talking about. Consider point 4 above to plot zoomed in results.

14.   Figure 5 is hard to view. why are there two green lines in the figure can you just plot a couple of profiles to highlight major idea of the plot ? What are the reasons for the differences of the shoreline regression rates ? Specially at the start of the simulation. You may not even need this plots because the general trend is similar.

15.   Line 175, Remove “the” significant. Also talk about the figure 6 a first. Follow some order

I think these plots can be made better again by splitting the idea of shoreline regression and sediment transport distance. Also marking them with the profile legends. Overall

Figure 6 is very hard to follow.

16.   Line 188: how is wave skewness greater than 1.0.  I understand that wave skewness can also be  defined as (ucrest/(ucrest-utrough)) which always leads to wave skewness < 1.0

17.  Figure 7. why for As=0.78 Sh=17.5 and for As=0.75 it is 15.5 m. You mentioned that more asymmetry leads to less Sh right ? If things are falling out of a pattern, please spend time explaining possible reasons.  

18.  Line 198: If you are referring to figure 3 then you should write the explanation over there. There needs to be a way out to make these figures more explanatory. Right now, you may need to improve figure 7 if you find a way to mention the asymmetry with a zoomed in version of figure 3 and then talk about the location of bar, depth, and shoreline regression on that. it will be much easy to understand then.

19.  Line 217: Conclusions; The second point of the conclusion is not reported in the results.

20.   Line 235 : Conclusion, Is there any literature supporting what you observe that higher wave asymmetry leads to decrease in coastal retreat.  

Author Response

(The authors gave the same response as above.)

Reviewer 3 Report

You can find all my comments in the attached file

Author Response

(The authors gave the same response as above.)

Round 2

Reviewer 2 Report

The authors have made a good attempt to improve the paper based on previous review comments. However, work needs to be done to improve the quality of manuscript as far as getting it edited thoroughly. Some of the new references are not in the format of JMSE requirements. Some of the newly added text also falls out of context at the end of the paper before the conclusions. May be there can be a separate section of discussion for that. Please edit the paper carefully. There are still several typos. 

Here are the specific issues. 

Line 17: Low frequency waves and mean wave period ? Correct this sentence perhaps

by removing mean wave period ->

Line 17 - 18. The two lines are very confusing and can be cleaned up by splitting them

into two separate sentences. The shoreline recession depends on height of low frequency

waves.. Then talk about what caused smaller retreat of shoreline. 

Line 31 till it joins*

Line 32 Increase of wave height can be replaced with stronger waves. Because you are still

talking about weak or moderate waves in the previous sentence. May be you should clarify

how you classify weak, moderate or strong waves. 

Line 45 protection

Line 67 Contradiction 

Line 68 resolves * 

Line 68 --> Good that you added this for clarification. 

Line 79 of a set of 

Line 92 D50 (d50) 

Line 95 overestimates 

Line 97 what is instationary ? 

Figure 3 Highly recommend using legends --> The quality of figure is bad.   

Caption -- > significant wave height ..also is the period, peak wave period  ?

Line 110 -> was set to 

Line 126 -> Why do you have the same formula written twice for mean wave period

Line 149 - Table remove additional "s" 

Line 169 --> Control

Line 171 --> Correct the english

Line 190 --> figure 6 --> spellings of profile 

Line 208-209- -> The smaller the change in the mean wave period 

line 225-> IGW affects coasts in 2 different ways:

a) bound long waves protect shore

B) break point....  

Line 254-271 -> I think the authors have added new text

to address concerns of another review. Also from 266-271

it is hard to follow the underlining point of all the explanation.

Overall these lines should be in a context. They are not very clear

On line 264 there is a typo u2m. 

for this paper after.

Line 269 is not clear. 

In my opinion even if the authors take it out, it is not going to affect

the quality of the paper. These things may find a context in a "DISCUSSION"

section that can separately written down. 

Conclusions

4. Line 300 directly related to the "change" in significant wave height across the bar.

Author Response

Many thanks to Reviewer 2

We appreciate all your remarks and try to correct all of them. 

Some troubles we have with Figure 3 only. Unfortunately it's impossible to change quality. We clarified wave parameters description. 

We get extra English check by our English-speaking colleague and hope now it's better.

Reviewer 3 Report

The reviewer want to thank you for the effort in modifying the manuscript

The revised version is clearly improved. All the comments of the reviewers have been clarified and the paper is now suitable for publication.

Author Response

Many thanks to Reviewer 2

We appreciate your work and time!

We get extra English check by our English-speaking colleague and hope now it's better.